# Laser-induced phase separation of silicon carbide

Insung Choi[1,2,*], Hu Young Jeong[3,4,*], Hyeyoung Shin[5], Gyeongwon Kang[5], Myunghwan Byun[1], Hyungjun Kim[5], Adrian M. Chitu[6], James S. Im[6], Rodney S. Ruoff[7,8], Sung-Yool Choi[2] & Keon Jae Lee[1,7]

Understanding the phase separation mechanism of solid-state binary compounds induced by laser–material interaction is a challenge because of the complexity of the compound materials and short processing times. Here we present xenon chloride excimer laser-induced melt-mediated phase separation and surface reconstruction of single-crystal silicon carbide and study this process by high-resolution transmission electron microscopy and a time-resolved reflectance method. A single-pulse laser irradiation triggers melting of the silicon carbide surface, resulting in a phase separation into a disordered carbon layer with partially graphitic domains ($\sim$2.5 nm) and polycrystalline silicon ($\sim$5 nm). Additional pulse irradiations cause sublimation of only the separated silicon element and subsequent transformation of the disordered carbon layer into multilayer graphene. The results demonstrate viability of synthesizing ultra-thin nanomaterials by the decomposition of a binary system.

[1] Department of Materials Science and Engineering, KAIST, Daejeon 34141, Republic of Korea. [2] School of Electrical Engineering, Graphene/2D Materials Research Center, Center for Advanced Materials Discovery for 3D Display, KAIST, Daejeon 34141, Republic of Korea. [3] UNIST Central Research Facilities (UCRF), UNIST, Ulsan 44919, Republic of Korea. [4] School of Materials Science and Engineering, UNIST, Ulsan 44919, Republic of Korea. [5] Graduate School of Energy, Environment, Water, and Sustainability (EEWS), KAIST, Daejeon 34141, Republic of Korea. [6] Program in Materials Science and Engineering, Department of Applied Physics and Applied Mathematics, Columbia University, New York, New York 10027, USA. [7] Center for Multidimensional Carbon Materials, Institute for Basic Science (IBS), Ulsan 44919, Republic of Korea. [8] Department of Chemistry, UNIST, Ulsan 44919, Republic of Korea. * These authors contributed equally to this work. Correspondence and requests for materials should be addressed to S.-Y.C. (email: sungyool.choi@kaist.ac.kr) or to K.J.L. (email: keonlee@kaist.ac.kr).

Laser beam-induced processing of materials has been utilized for tuning material properties due to the ability to rapidly heat to the melting point and it allows the controlled surface modification of materials[1–3]. Laser thermal processing has been used to activate dopants introduced by ion implantation to modify the intrinsic properties of semiconductor materials such as Si and Ge to form ultra-shallow junctions[4–6]. Furthermore, laser beam-induced melting and solidification aids the fabrication of low-temperature polycrystalline silicon on glass or flexible substrates, which is well-established in producing thin film transistors for displays[7,8]. Though many fundamental research studies related to an understanding of laser-induced phase transformations in elements have been made[9–12], laser interactions with binary compounds have been less studied because of the relative complexity of the compound materials[13–15].

The thermal decomposition of silicon carbide (SiC) has demonstrated a possibility for the direct synthesis of high-quality graphene on an insulating substrate[16–18]. However, the extremely high temperature ($\sim 2,000$ K) needed for conventional furnace processing limits its compatibility with industrial semiconductor applications. In this respect, excimer laser irradiation can be an alternative technique for the thermally driven surface reconstruction of SiC. Recently, several research groups have reported a sublimation of Si atoms on a SiC surface under nanosecond pulsed laser irradiation for the synthesis of graphene[19–21]. However, an understanding of the graphitization mechanism has not been achieved due to the difficulty of observing the time sequence of the laser-induced decomposition of the binary compound into each elemental material.

Here we demonstrate the melt-mediated phase separation and surface reconstruction of single-crystal 4H-SiC by irradiation of a xenon chloride (XeCl) excimer laser ($\lambda = 308$ nm, pulse duration $\sim 30$ ns), resulting in formation of ultra-thin elemental material. Time-resolved reflectance (TRR) analysis shows an explicit information on the melting and solidification of 4H-SiC surface by laser irradiation[22,23]. A single-pulse irradiation of laser fluence (that is, laser energy per unit area, $1,653$ mJ cm$^{-2}$) allows the 4H-SiC surface to be melted and leads to the phase separation of liquid SiC into a disordered carbon (C) layer with graphitic domains ($\sim 2.5$ nm) on a polycrystalline silicon (poly-Si, $\sim 5$ nm) layer in nanoseconds. Through molecular dynamics (MD) simulations, we confirm that graphitic C had the lowest surface potential energy in Si-C binary systems[24,25]. Irradiation with a second pulse causes sublimation of only the separated Si, while the disordered C layer is transformed into a layered structure with increased crystallinity. These results are systematically investigated by high-resolution transmission electron microscopy (HRTEM), demonstrating that additional pulse irradiations lead to a gradual increase in the thickness of C layer, eventually to form multilayer graphene. Our study indicates the potential for nanomaterial synthesis through the phase separation of a binary compound by laser-induced melting and solidification.

## Results

### Cross-sectional TEM analysis of phase separation.
To examine the phase separation and surface reconstruction of single-crystal 4H-SiC with laser irradiation, cross-sectional TEM observations were made. A platinum (Pt) thin layer was deposited to protect the top surface before sample preparation using focused ion beam milling. Figure 1a is a bright-field TEM image of a 4H-SiC surface irradiated by a single pulse from the laser, showing distinguishable layers formed on the original substrate (see details in Methods). To clearly identify these regions, we acquired HRTEM images using a spherical aberration-corrected TEM operated at 80 kV. A HRTEM image (Fig. 1b) shows that the

brighter region of Fig. 1a is composed of two different phases (that is, C ($\sim 2.5$ nm) and Si layers ($\sim 5$ nm)); the former is referred as a randomly stacked, layered C structure and the latter is poly-Si with clear lattice fringes over a local area. A fast Fourier transform pattern shows that there is a 3C-SiC layer with the A–B–C stacking sequence between the poly-Si layer and the 4H-SiC surface, (Supplementary Fig. 1). In Fig. 1c, the poly-Si phase is confirmed by a high-magnification HRTEM image and fast Fourier transform patterns with (111) and (220) reflections. The top surface layer is observed to be disordered C with partially graphitic domains. To identify the C and Si layers, we used an energy-filtered TEM (EFTEM) for elemental mapping analysis (Fig. 1d). C mapping shows that the top layer on the surface consists of only C element without Si detection. In contrast, Si mapping shows the formation of a Si layer between the C and the SiC substrate, thereby demonstrating phase separation into C and Si layers. The atomic volume ratio of graphitic C to cubic Si was calculated to be $\sim 1:2$ (see Methods) corresponding to the thickness ratio of C ($\sim 2.5$ nm) and Si layers ($\sim 5$ nm), as shown in Fig. 1b. In addition, a HRTEM image (Fig. 1e) of the interfacial region between 3C and 4H-SiC shows that (111)-oriented 3C-SiC was epitaxially grown on (0001)-oriented 4H-SiC.

### MD simulations for phase separation process.
In an attempt to theoretically understand the origin of the melt-mediated phase separation of SiC, MD simulations were performed. To capture the key intermediates during the melting and solidification, which can be hardly sampled from a brute-force MD simulation, we manually built structures corresponding to the initial, intermediate and final states over the course of phase separation (full details about the structures are given in the Methods section). Each state of them was equilibrated by performing a MD simulation, and then minimized to obtain the energy. We considered eight different intermediate states connecting from the initial solid state of SiC (SiC ($s$), Supplementary Fig. 2) to the final state, where most C atoms are graphitized from the liquid phase of SiC (SiC ($l$)).

We found that the SiC ($s$) has the most stable energy, while the intermediate state where no carbon is yet graphitized from SiC ($l$) has the most unstable energy. Although the energy becomes gradually lowered as the graphitization process proceeds, multilayer graphene of the final state is still unstable compared with the SiC ($s$), as schematically shown in Fig. 2 (for details see Supplementary Fig. 3). This illustrates the energetic changes accompanied with the phase separation process; laser irradiation on the SiC ($s$) leads to the formation of highly unstable SiC ($l$) due to the heating up to $\sim 3,500$ K (Supplementary Fig. 4). This triggers the system to reach a metastable state with the formation of multilayer graphene. In addition, we found that graphitic C layer has the lowest surface energy in the Si-C binary system from an analysis of the atomic potential energy spectra of the systems (Supplementary Fig. 5), which can be understood as the driving force for the graphitization process. From these theoretical investigations, we conceive that the highly non-equilibrium character of the laser thermal processing helps the process be kinetically controlled, inducing a phase transform to the metastable state.

### TRR analysis for melt-mediated phase transformation.
The TRR analysis was used to identify the melt-mediated transformation by using the different reflectivities of the solid and liquid phases (see details in Methods and Supplementary Fig. 6). The inset in Fig. 3 shows an optical microscope image of the laser beam-irradiated area, $1 \times 1$ mm$^2$, which was used to measure the TRR signal. The TRR signal (Fig. 3) is increased from 50 to 75 ns by supplying the second hump (peak) of the laser pulse, indicating melting of 4H-SiC surface. The unusually high

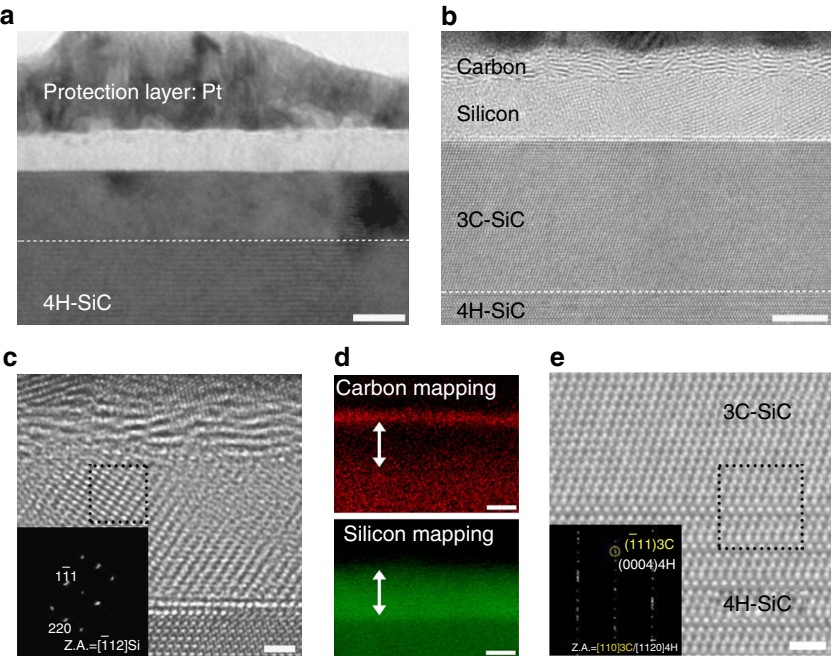

**Figure 1 | Phase separation of 4H-SiC by XeCl excimer laser irradiation.** (**a**) Bright-field TEM image of 4H-SiC surface with single-pulse irradiation. The dotted line indicates the boundary between newly formed layers and the 4H-SiC substrate. Scale bar, 10 nm. (**b**) HRTEM image of a phase separated area and a 3C-SiC layer on top of the 4H-SiC in **a**. Scale bar, 5 nm. (**c**) Magnified HRTEM image of the poly-Si layer. Scale bar, 1nm. Inset shows a fast Fourier transform pattern of the black-dotted area. (**d**) EFTEM mapping of C and Si elements of the phase-separated layer. Double-sided arrows indicate the position of the poly-Si layer. Scale bars, 5 nm. (**e**) Magnified HRTEM image of the boundary area between 3C-SiC and 4H-SiC. Inset is the FFT pattern of the interface between 3C-SiC and 4H-SiC in the black-dotted area. Scale bar, 1nm.

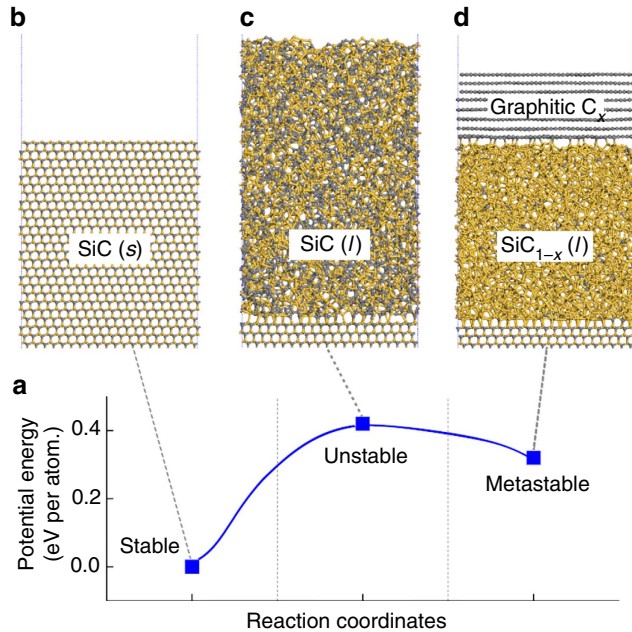

**Figure 2 | MD simulations for phase separation process.** (**a**) Potential energy plot based on the MD simulations for the melt-mediated phase separation. The blue solid line indicates schematic of potential energy variation depending on the laser-induced melting and solidification process of SiC surface. The schematic figures show side view of each step such as (**b**) SiC (*s*), (**c**) SiC (*l*)/SiC (*s*) and (**d**) graphitic $C_x$/SiC$_{1-x}$ (*l*)/SiC (*s*) systems taken from the MD simulations. Grey and yellow spheres represent C and Si atoms, respectively.

intensity at 75 ns is estimated to include a reflectance signal from the 4H-SiC surface and laser signal (see Supplementary Note 1 for details). A plateau region between 180 and 450 ns indicates the presence of a liquid Si layer on the surface. This plateau region strongly suggests that the phase separation occurred within 180 ns (see Supplementary Note 1 for details).

**Effect of an additional irradiation pulse.** We examined a 4H-SiC surface irradiated with an additional pulse (the second, 1 Hz of laser pulse frequency) to obtain additional phase separation and reconstruction through a melt-mediated phase transformation. A bright-field TEM image (Fig. 4a) shows a thin layer with bright contrast and the formation of a 3C-SiC layer on 4H-SiC. Compared with single-pulse irradiation, a HRTEM image (Fig. 4b) of the top surface in Fig. 4a indicates the absence of a poly-Si layer and a graphitic C layer with increased crystallinity. Electron energy loss spectroscopy (EELS) analyses were performed to compare the crystallinity of the two samples (Fig. 4c). The C layer formed by the second irradiation pulse had increased crystallinity with a sharper $\sigma^*$ peak compared with that produced by single-pulse irradiation. The local non-uniform area on the 4H-SiC surface (Supplementary Fig. 7a) was investigated with a HRTEM to explain how the poly-Si layer observed in Fig. 1b disappears. A high-magnification image (Supplementary Fig. 7b) of an island structure, stacked on the 3C-SiC surface, indicates a residual poly-Si layer covered by an extremely disordered C layer. Additional analyses with EFTEM elemental mapping using C K- and Si L-edges clearly shows some remaining Si (Supplementary Fig. 7c,d). From these observations, we propose that Si atoms under a strongly bonded C layer rarely escape through the C layer because of the extremely short annealing time. Therefore, the disappearance of elemental Si below the C layer can be attributed to the sublimation of the

poly-Si layer (melting point, $m_p \sim 1,700$ K) induced by the laser fluence of $\sim 1,653$ mJ cm$^{-2}$, which is high enough to melt the SiC ($m_p \sim 3,100$ K). In contrast, C material is remained on the surface due to its extremely high melting temperature ($m_p \sim 4,000$ K)[26].

Figure 4d shows the TRR signal of a surface that was irradiated with the additional pulse (the second) on the disordered C/poly-Si/3C-SiC/4H-SiC structure shown in Fig. 1b. The TRR signal is noticeably increased at 37 ns as soon as the first hump of the second laser pulse was incident on the surface. We assume that

the high intensity at 37 ns is due to a sublimation of the separated poly-Si layer. The flattened region between 150 and 290 ns clearly indicates liquid SiC (see Supplementary Note 2 for details). The inset in Fig. 4d compares the TRR signals of single and the second irradiation pulses. Note that the higher intensity and longer flattened region of the TRR signal, observed only in the single-pulse irradiation, are related to the generation of separated Si element.

**Investigation of multi-pulse irradiations.** To investigate the formation process of multilayer graphene, multi-pulse irradiations were carried out and characterized. Figure 5a,b shows HRTEM images of 4H-SiC surfaces after 3 and 10 irradiation pulses, respectively. Both clearly show no further phase separation and new surface reconstruction, compared with single or two irradiation pulses. From these observations, we believe that additional phase separation cannot be developed by multi-pulse irradiations because the C layer, which has the lowest surface energy, has already been formed by the first irradiation pulse. On the other hand, the 4H-SiC surface with 10 irradiation pulses (Fig. 5b) shows an increased thickness of the C layer, which corresponds to multilayer graphene, while the thickness of the 3C-SiC layer decreases. To verify the thickness changes in the C and 3C-SiC layers with the number of irradiation pulses, TEM analyses were performed for the samples irradiated with 1, 2, 3, 10, 30 and 100 pulses. Figure 5c shows the thicknesses of the C (blue line) and 3C-SiC layers (black line) as a function of the number of pulses. The thickness of the C layer ($\sim 2.2$ nm) formed by single-pulse irradiation was slightly decreased ($\sim 1.3$ nm) after the second irradiation pulse due to an improvement of the crystallinity with $sp^2$ bonding. The average thickness (square label) of the C layer increased gradually after the tenth pulse, while the 3C-SiC layer became thinner. Representative

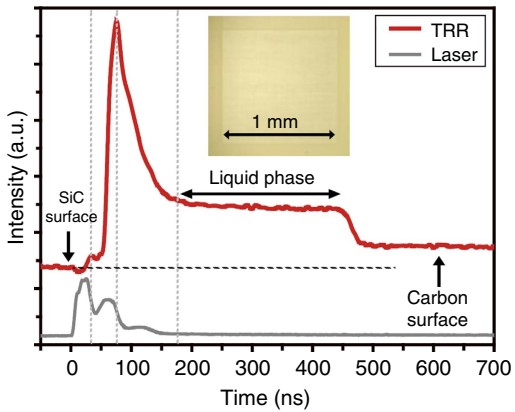

**Figure 3 | TRR signal of 4H-SiC surface with single-pulse irradiation.** TRR method is used to prove melt-mediated phase transformation by using different reflectances of the solid and liquid phases. The red line indicates reflectance trace of 4H-SiC surface before and after single-pulse irradiation. The grey line presents laser intensity as a function of time and the inset shows an optical microscope image of the laser irradiated surface, $1 \times 1$ mm$^2$.

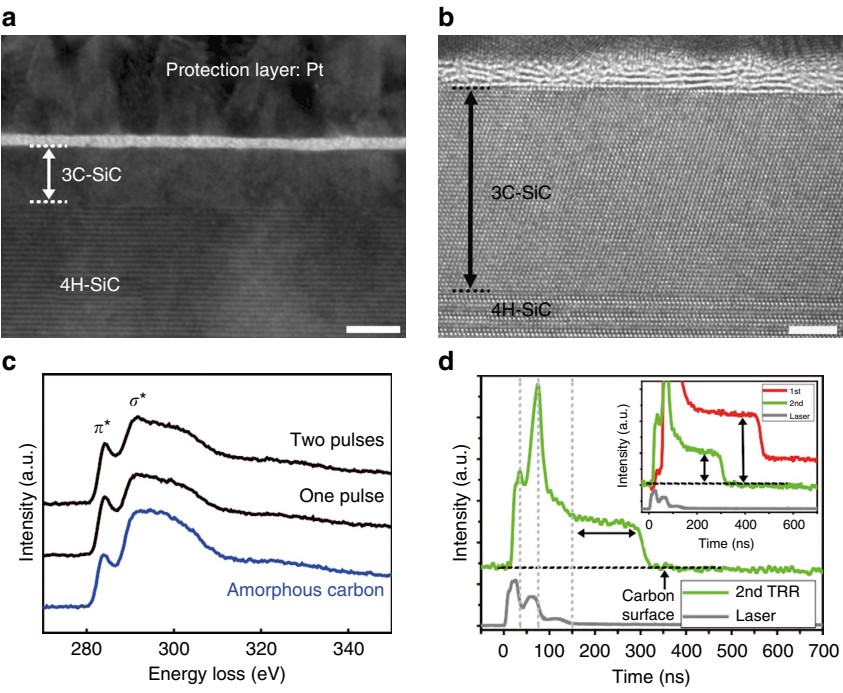

**Figure 4 | Sublimation of only the poly-Si layer from the phase-separated layer.** (**a**) Bright-field TEM image of 4H-SiC surface after two irradiation pulses. Scale bar, 10 nm. (**b**) HRTEM image of graphitic C and 3C-SiC layers on top surface in **a**. Scale bar, 3 nm. (**c**) EELS spectra of C layers (black lines) obtained from surfaces with one and two irradiation pulses. The blue line is the spectrum of an amorphous C layer as reference. (**d**) TRR signal of the 4H-SiC surface during the second irradiation pulse. The grey line shows the laser intensity as a function of time. Inset compares the absolute amount of reflectances produced by one irradiation pulse (the first, red line) and an additional pulse (the second, green line).

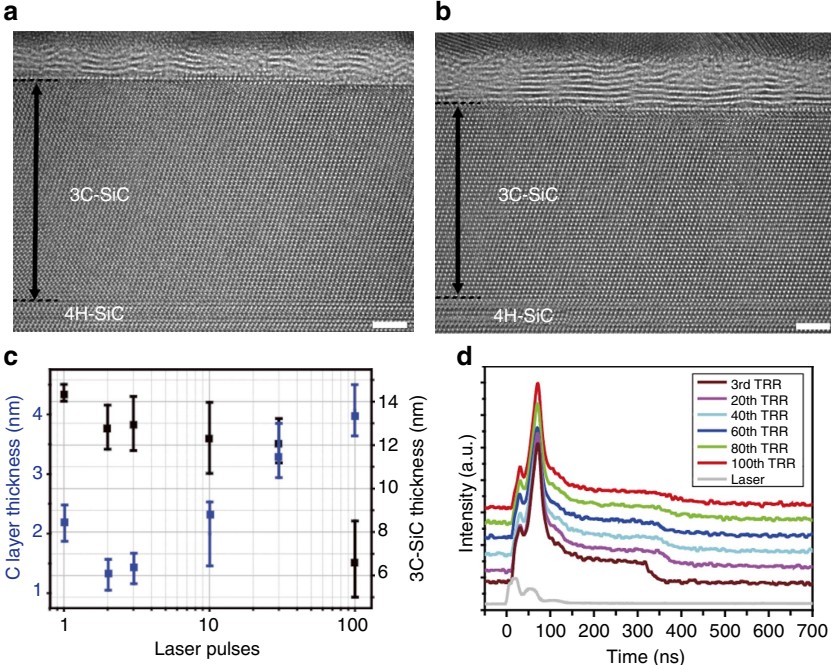

**Figure 5 | Increase in the number of C layers by multi-pulse irradiation.** (**a,b**) HRTEM images of 4H-SiC surfaces after a 3 (**a**, left) and 10 (**b**, right) irradiation pulses. Double-sided arrows indicate 3C-SiC layers. Scale bars, 2 nm. (**c**) Systematic investigation of thickness variations of the C and 3C-SiC layers as a function of the number of laser pulses from 1 to 100. Square labels represent the average values of thickness from each pulse. Error bars indicate minimum and maximum values of thickness at each pulse. (**d**) TRR signals of 4H-SiC surfaces with multi-pulse irradiations, from the third to the hundredth pulse.

HRTEM images and Raman data for laser irradiation up to 300 pulses are shown in Supplementary Fig. 8. Three representative peaks at Raman spectra, including the defect-induced D peak, in-plain vibrational G peak and two-phonon scattered 2D peak are clearly observed at 1,364, 1,583 and 2,720 cm$^{-1}$, respectively. By increasing the number of irradiation pulses, intensity of two-phonon scattered 2D peak is increased while D peak is decreased. EELS analyses were used to characterize the improvement in the crystallinity of the C layer (Supplementary Fig. 9). The EELS spectrum for 100 pulses is comparable to that of chemical vapour deposition-grown multilayer graphene with a random stacking. In addition, the TRR signals for 4H-SiC surface with many irradiation pulses (Fig. 5d) were examined to characterize the melting and solidification process. All TRR data show a flat region that indicates melt-mediated decomposition and solidification of the 4H-SiC surface for the formation of multilayer graphene (see Supplementary Note 3 for details).

## Discussion

We have investigated XeCl excimer laser–SiC interaction to understand decomposition mechanism of solid-state binary compound by using a wide range of laser fluences and various number of irradiation pulses. Supplementary Fig. 10 shows HRTEM images of 4H-SiC surfaces after single and multi-irradiation of laser fluence E1, E2 and E3 corresponding to 1,000, 1,200 and 1,653 mJ cm$^{-2}$, respectively. We found two major regimes of amorphization and phase separation by single-pulse irradiation, which strongly depend on laser fluence. Low laser fluences of E1 and E2 induced formation of thin amorphous layers, such as ~3 and ~17 nm (Supplementary Fig. 10a,b), through melting and quenching process[9]. To investigate the relationship between the number of laser pulses and formation of graphitic C layer, multi-irradiation pulses with E1 and E2 were performed. Although many pulses up to 600 were irradiated on 4H-SiC surface, no graphitized C layer was observed at the laser fluence E1

(Supplementary Fig. 10d,e). In contrast, laser fluence E2 caused graphitization from the amorphous SiC layer (~17 nm) through sublimation of Si element (Supplementary Fig. 10f), C nucleation (Supplementary Fig. 10g), formation of multilayer graphene (Supplementary Fig. 10h), in sequence, by increasing the number of laser pulses up to several hundreds of irradiations. On the other hand, thin phase-separated C and Si layers were observed by using high laser fluence of E3 (Supplementary Fig. 10c).

From the experimental results, we have found required conditions for the phase separation as followings: (1) solid-state binary material should include one element that has the lowest surface energy from liquid state of binary system. (2) The one element should have a larger melting temperature than both the other element and binary compound. For this reason, compound material, which includes carbon, has a high possibility for a phase separation. (3) A wide bandgap compound materials are good candidates due to their characteristic of semi-insulator. (4) We have to avoid melt-mediated amorphous phase transition via quenching, instead of the phase separation, which is mainly related to pulse duration and energy density of laser.

In conclusion, we demonstrate that a single-pulse irradiation induces melt-mediated phase separation into C and Si layers on SiC surface by characterization with HRTEM and TRR analysis. MD simulations indicate that the graphitic C layer is formed by minimization of surface energy from liquid SiC. Additional pulses lead to the selective sublimation of elemental Si from the phase-separated layers due to the different melting temperatures of the two elements. Therefore, our results give indication of synthesizing ultra-thin nanomaterials through the laser-induced phase separation of a binary system.

## Methods

**Sample preparation and laser irradiation.** 4H-SiC wafers with low doping ($n = 2.2 \times 10^{18}$ cm$^{-3}$) and chemical mechanical polishing were purchased from Cree. Specimens, $5 \times 6$ mm$^2$, were cut from the wafers and cleaned by sequential ultrasonic baths in acetone, isopropyl alcohol and deionized water to remove

grease. After cleaning, a SiC substrate was placed into a heating stage (Linkam, TS1500). This is a small chamber ($100 \times 100$ mm²) that enables the bottom of the substrate to be heated up to 1,000 °C with inert gas supply. The sample was annealed at 650 °C for surface cleaning and it was maintained at the same temperature during laser irradiation in an Ar flow. The laser system used is comprised of a XeCl excimer laser (Coherent, LPX model, $\lambda = 308$ nm, pulse duration ~30 ns) and beam delivery optics with a homogenizer optical system. The laser beam is concentrated $5\times$ by a projection lens through a square pattern mask. Each pulse had a laser fluence of 1,653 mJ cm$^{-2}$ and a repetition rate of 1 Hz. For multi-pulse irradiations over 100 pulses, 5 Hz was used to reduce the processing time. Both repetition rates appeared to have the same effect because the heating and cooling produced by the nanosecond pulse are completed within a microsecond. All the laser experiments were performed with the same substrate and experimental conditions except for the number of pulses.

**TEM analysis.** To prepare cross-sectional TEM samples of the 4H-SiC, we used a focused ion beam milling technique (FEI, Helios 450HP). A Pt metal layer was deposited to protect the top surface of the sample before milling. To make much thinner and undamaged TEM samples, we additionally milled the samples using a low-energy Ar-ion milling system (Fischione Model 1040 Nanomill). To structurally characterize the top surface, we acquired atomic resolution TEM images using a spherical aberration-corrected TEM (FEI Titan3 G2 60-300 with an image-forming Cs corrector) operated at 80 kV, because C-based materials can be easily damaged by high-energy electron beam irradiations. EFTEM elemental mapping and EELS spectra were recorded using a Gatan Imaging Filter (Gatan Quantum 965 ER). EELS spectra were collected in the STEM mode using a dispersion of 0.25 eV per channel and a 2.5 mm aperture. The elemental mapping was acquired with the three-window method.

**Numerical simulation.** The numerical simulation performed in this study is widely used for the analysis of the thermal behaviour of laser–solid interactions involving heat generation and cooling at the surface of the irradiated area by light absorption[27]. Thermal simulation parameters of SiC were utilized with $\{720/(T-69)\}$ of thermal conductivity and $\{1.12 \times \ln(T) - 4\}$ of heat capacity with reference to previous studies[21]. In addition, values of the real part $n$ (2.9) and imaginary part $k$ (0.1) of the refractive index as optical parameters were used for a 308 nm wavelength[21]. The laser intensity as a function of time of our 308 nm XeCl laser (Supplementary Fig. 6b) and optimized experimental conditions (that is, fluence and substrate heating) were used to investigate the temperature history of the SiC surface during a short pulse (30 ns).

**Measurement of TRR signal and laser intensity as a function of time.** The TRR system is illustrated in Supplementary Fig. 6a. It consists of two photodetectors (Thorlabs, DET10A), a filter (Thorlabs, FEL0500), a probing laser diode (635 nm) and an oscilloscope (Tektronix, TDS3054). The TRR method is used to detect changes in the laser diode signal during 308 nm laser irradiation of the sample surface because semiconductor materials such as Si and SiC have a higher reflectance in the liquid phase that is maintained longer than the time of the laser pulse[22,23]. For this reason, the principle of TRR analysis uses different reflectances of materials in the solid and liquid phases. One photodetector is for the analysis of the reflectance from the SiC surface. Generally, the photodetector can monitor a wide range of wavelengths, from 200 to 1,100 nm. Therefore, a filter is combined with the photodetector and serves the important role of blocking the signal of the 308 nm laser because we are interested in only information from the sample surface (here, liquid phase of SiC). The other photodetector detects the 308 nm signal of the laser beam. The two detectors are synchronized with an oscilloscope.

The 308 nm XeCl laser has a unique intensity variation as a function of time with three humps (peaks), which were measured by a photodetector with an oscilloscope (Supplementary Fig. 6b). Normally, the pulse duration (full width of half maximum) is considered to be ~30 ns, which is only for the first hump. In this study, the second hump causes accumulation of thermal energy with the first hump, resulting in the highest surface temperature (Supplementary Fig. 4) and reflectance signals (Figs 3 and 4d) at 75 ns.

**Calculation of atomic volume ratio of Si and C.** To evaluate the atomic volume ratio of Si to C, we carried out density functional theory (DFT) calculations using the Vienna *Ab-initio* Software Package programme[28]. All DFT computations were performed using a plane-wave basis set and the Perdew–Burke–Ernzerhof exchange-correlation functional coupled with an empirical van der Waals correction (PBE-ulg) to provide a good description of the London dispersion forces between C and Si atoms[29]. Monkhorst $k$-point grids of ($11 \times 11 \times 11$) and ($14 \times 28 \times 10$) were used for the Si crystal and graphite unit cells, respectively, and an energy cutoff of 600 eV was considered for the plane-wave basis set.

After optimizing the unit cells of cubic Si with the diamond structure and graphite, we found that the atomic volume ratio is ~1:2 for graphitic C and cubic Si (8.75 versus 19.75 Å³ per atom). All details for the optimized structures, including the number of atoms, lattice parameters and volumes are listed in Supplementary Table 1 (ref. 28).

**MD simulation.** It has been widely discussed that the direct simulation of the solidification process is hardly available using conventional MD simulations[30,31] while the simulation of melting process is possible. Also, it is still ambiguous how the laser heating effect can be incorporated in MD simulations, albeit there are several attempts[32,33]. Instead of performing a direct MD simulation for the overall process, therefore, we built 10 different structures modelling the initial, intermediate and final states of the phase separation process: (Model 1) initial solid state of SiC (SiC (*s*)) is built using a slab model of a Si-terminated 3C-SiC (111) slab model (Supplementary Fig. 3), because the graphitic layers are observed to be grown on top of 3C-SiC (111) layers. The lateral dimension of the slab model is a $5\sqrt{6}a_0 \times 3\sqrt{2}a_0$ unit cell, where the lattice parameter of $a_0$ is 4.34 Å (chosen from DFT result), consisting of 40 bilayers (4,800 Si + 4,800 C atoms). (Model 2) To model the local heating effect of laser irradiation, top 25 bilayers of the surface of Model 1 were assumed to be melted. We consider a system consisting of only the top 25 bilayers of SiC (*s*) in the simulation cell where periodic boundary condition is applied, namely Model 1-top (3,000 Si + 3,000 C atoms). Using a separate MD simulation of Model 1-top, we sampled the liquid phase structure of SiC (*l*) at high temperature of 3,100 K for 2–3 ns. After removing the top 25 bilayers from Model 1 of SiC (*s*), the sampled structure of SiC (*l*) is located. (Models 3–10) Considering the lateral dimension of the simulation cell for the slab model, one layer of the graphene is required to be modelled using 320 C atoms. To model the formation of *n*-layer graphene from SiC (*l*), we randomly removed ($320 \times n$) C atoms from Model 1-top, which is simulated under high temperature of 3,100 K to sample liquid structures of SiC$_{1-x}$ (*l*) ($x = 0.107 n$). After removing the top 25 bilayers from Model 1, the sampled structure of SiC$_{1-x}$ (*l*) is located, and then *n*-layer graphene is located on top of SiC$_{1-x}$ (*l*). Models 3–10 correspond to $n = 1$–8 (final state).

For above models, MD simulations were performed using the large-scale atomic modelling massively parallelized simulation code[24] at 300 K for 10–20 ns. The interatomic interactions is described using Tersoff potential[25], time step is set as 1 fs and a Nosé–Hoover thermostat (damping constant = 100 fs) was used for thermostating.

**Data availability.** The data that support the findings of this study are available from the corresponding author on request.

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

## Acknowledgements

K.J.L. and R.S.R. acknowledge that this work was supported by IBS-R019-D1. S.-Y.C. acknowledges the support from Creative Materials Discovery Program (NRF-2016M3D1A1900035).

## Author contributions

I.C., H.Y.J., S.-Y.C. and K.J.L. designed the project and analysed the whole data; I.C. performed the laser experiments, analysed TRR signals, performed numerical simulation and obtained Raman data; H.Y.J. performed TEM and EELS analyses; H.S., G.K. and H.K. performed MD simulations, DFT simulations and calculation of atomic volume ratio; A.M.C. carried out measurement of TRR signals; J.S.I. contributed to the numerical simulation and analysis of melt-mediated phase separation; I.C., H.Y.J., H.S., G.K., H.K., S.-Y.C. and K.J.L. co-wrote the manuscript; M.B. and R.S.R. discussed results and revised the manuscript; K.J.L. and S.-Y.C. are responsible for managing all aspects of this study including the writing of manuscript.

## Additional information

**Competing financial interests:** The authors declare no competing financial interests.

