## [Peer Review File · Nature Communications]

PEER REVIEW FILE

Reviewers' comments:

Reviewer #1 (Remarks to the Author):

I read the prior reviews and the authors' responses and manuscript revisions. Not much more needs to be said. The manuscript contains significant original results that would be of interest to a broad community of researchers.

Reviewer #3 (Remarks to the Author):

The authors present a revised version of their manuscript originally submitted to Nature for consideration in Nature Communications. The improvements do make the manuscript better and Nature Communications offers an adequate platform for the publication of these results. I do not see any major reasons why the article should not be published however I would like the authors to answer my two following concerns/questions. The authors call their material "multi-layer graphene", how is this different from graphite? Have they investigated the stacking (Bernal vs. random)? It is also peculiar that Raman spectroscopy of the C-rich layer is not discussed in the manuscript (as far as I can tell, the only Raman spectra pertain to the SiC in the SI). Raman is a widely available and accepted technique to determine the quality of the C-layer by looking at the amplitude of the defect peak vs. the amplitude of the 2D peak. The SiC background from the spectra in the SI could be subtracted to determine the spectrum of the carbon layer and make a statement on the perfection of the C-rich layer.

Response Letter

[Reviewer #1]

Overall comments: *“I read the prior reviews and the authors' responses and manuscript revisions. Not much more needs to be said. The manuscript contains significant original results that would be of interest to a broad community of researchers.”*

Our response: We deeply appreciate the reviewer's high evaluation of our manuscript.

[Reviewer #3]

Overall comments: *“The authors present a revised version of their manuscript originally submitted to Nature for consideration in Nature Communications. The improvements do make the manuscript better and Nature Communications offers an adequate platform for the publication of these results. I do not see any major reasons why the article should not be published however I would like the authors to answer my two following concerns/questions.”*

Our response: We appreciate reviewer's evaluation of our manuscript. Our responses to the comments are described as below.

Comment 1: *“The authors call their material "multi-layer graphene", how is this different from graphite? Have they investigated the stacking (Bernal vs. random)?”*

Our response: We used “multilayer” to express few-layer graphene. Figure R1 compares EELS spectra of laser-induced multilayer graphene, CVD-grown multilayer graphene, and graphite as a reference. The graphite with a Bernal (A-B) stacking shows a high intensity of π^* . On the contrary, EELS spectrum of laser-induced multilayer graphene is comparable to that of CVD-grown multilayer graphene, showing a similar π^*/σ^* ratio. Therefore, we noted that “The EELS spectrum for 100 pulses is comparable to that of CVD-grown multilayer graphene” in the manuscript.

Previous studies of Lee and our group reported that laser-induced multilayer graphene are not Bernal-stacked and electronically decoupled through Raman analysis^{19,21}. The 2D peak of the laser-induced multilayer graphene on Si-terminated surface of SiC was in a good harmony with single Lorentzian. Single-peak fitting of the 2D peak normally observed in multilayer epitaxial graphene on the C-terminated surface of SiC by thermal decomposition in furnace, providing a sign of electronic decoupling between the graphene layers (Ref. R1). In terms of melt-mediated phase separation which is driven by minimization of surface energy, we believe that random stacking is more reasonable in our experimental condition.

Figure R1 | Comparison of EELS spectra of multilayer graphene produced by 100 irradiation pulses, CVD-grown multilayer graphene, and graphite.

Modification to the manuscript:

The text has been modified to make this statement more clear: “**The EELS spectrum for 100 pulses is comparable to that of CVD-grown multilayer graphene with a random stacking.**” in the revised manuscript (on page 9, line 8-9).

Comment 2: “*It is also peculiar that Raman spectroscopy of the C-rich layer is not discussed in the manuscript (as far as I can tell, the only Raman spectra pertain to the SiC in the SI). Raman is a widely available and accepted technique to determine the quality of the C-layer by looking at the amplitude of the defect peak vs. the amplitude of the 2D peak. The SiC background from the spectra in the SI could be subtracted to determine the spectrum of the carbon layer and make a statement on the perfection of the C-rich layer.*”

Our response: We appreciate the reviewer’s important comments and suggestion on Raman analysis. In response to reviewer’s comments, we subtracted SiC background from original Raman data (Ref. R2 and R3) and obtained concrete information regarding quality of carbon layer, as shown in Figure R2. A strong D peak at 1364 cm^{-1} indicates the presence of many defects and grain boundaries. By increasing the number of irradiation pulses from 100 to 300 pulses, intensity of two phonon scattered 2D peak is increased while D peak is decreased. We believe that quality of material can be improved by control of various experimental conditions such as laser wavelength (absorbance), pulse duration, cooling rate, gas condition, and pressure.

Figure R2 | Comparison of Raman spectra of laser-induced multilayer graphene on 4H-SiC. The black, red, and green lines correspond to 100, 200, and 300 irradiation pulses, respectively. The spectra of the D (1364 cm^{-1}) and 2D peak (2720 cm^{-1}) shown here are corrected by subtraction of a 4H-SiC background (Ref. R2 and R3).

Modification to the manuscript:

The text has been included to make this statement more clear: “Three representative peaks at Raman spectra, including the defect-induced D peak, in-plane vibrational G peak, and two phonon scattered 2D peak are clearly observed at 1364 , 1583 , and 2720 cm^{-1} , respectively. By increasing the number of irradiation pulses, intensity of two phonon scattered 2D peak is increased while D peak is decreased.” in the revised manuscript (on page 9, line 3-6).

Modification to the supplementary information:

Taking the reviewer’s helpful comments, we modified Supplementary Figure 8 (d) in the revised Supplementary Information.

Supplementary Figure 8 | (d) Raman spectra of laser-induced multilayer graphene on 4H-SiC surfaces after irradiation of 30 (black line), 100 (red line), 200 (blue line), and 300 pulses (green line). Three representative peaks, including the defect-induced D peak, in-plane vibrational G peak, and two phonon scattered 2D peak are clearly observed at 1364, 1583, and 2720 cm^{-1} , respectively. The inset shows magnification of 2D peak. The gray line is the Raman spectrum for an original 4H-SiC substrate as a reference.

References of Response Letter

- R1. Faugeras, C. *et al.* Few-layer graphene on SiC, pyrolytic graphite, and graphene: A Raman scattering study. *Appl. Phys. Lett.* **92**, 011914 (2008).
- R2. Emtsev, K. V. *et al.* Towards wafer-size graphene layers by atmospheric pressure graphitization of silicon carbide. *Nature Mater.* **8**, 203 (2009).
- R3. Röhrl, J. *et al.* Raman spectra of epitaxial graphene on SiC(0001). *Appl. Phys. Lett.* **92**, 201918 (2008).

Reviewers' Comments:

Reviewer #2 (Remarks to the Author):

The authors addressed my comments thoroughly and competently: as far as I am concerned the manuscript can be published in Nature Communications.

Response Letter

[Reviewer #2]

Overall comments: *“The authors addressed my comments thoroughly and competently: as far as I am concerned the manuscript can be published in Nature Communications.”*

Our response: We deeply appreciate the reviewer’s high evaluation of our manuscript.